# Association between the Concentrations of Metallic Elements in Maternal Blood during Pregnancy and Prevalence of Abdominal Congenital Malformations: The Japan Environment and Children’s Study

**DOI:** 10.3390/ijerph181910103

**Published:** 2021-09-26

**Authors:** Chihiro Miyashita, Yasuaki Saijo, Yoshiya Ito, Atsuko Ikeda-Araki, Sachiko Itoh, Keiko Yamazaki, Sumitaka Kobayashi, Yu Ait Bamai, Hideyuki Masuda, Naomi Tamura, Mariko Itoh, Takeshi Yamaguchi, Shin Yamazaki, Reiko Kishi

**Affiliations:** 1Center for Environmental and Health Sciences, Hokkaido University, North-12, West-7, Kita-ku, Sapporo 060-0812, Japan; miyasita@med.hokudai.ac.jp (C.M.); AAraki@cehs.hokudai.ac.jp (A.I.-A.); vzbghjn@den.hokudai.ac.jp (S.I.); kyamazaki@cehs.hokudai.ac.jp (K.Y.); sukobayashi@cehs.hokudai.ac.jp (S.K.); u-aitbamai@med.hokudai.ac.jp (Y.A.B.); hmasuda@cehs.hokudai.ac.jp (H.M.); ntamura@cehs.hokudai.ac.jp (N.T.); mitoh@cehs.hokudai.ac.jp (M.I.); takeshi7698@med.hokudai.ac.jp (T.Y.); 2Department of Social Medicine, Asahikawa Medical University, Midorigaoka-higashi 2-1-1-1, Asahikawa 078-8510, Japan; y-saijo@asahikawa-med.ac.jp; 3Faculty of Clinical Medicine, Japanese Red Cross Hokkaido College of Nursing, 664-1 Akebono-cho, Kitami 090-0011, Japan; yoshiya.ito@gmail.com; 4Faculty of Health Sciences, Hokkaido University, North-12, West-5, Kita-ku, Sapporo 060-0812, Japan; 5National Center for the Japan Environment and Children’s Study, National Institute for Environmental Sciences, 16-2 Onogawa, Tsukuba 305-8506, Japan; yamazaki.shin@nies.go.jp

**Keywords:** metallic elements, abdominal congenital malformations, prenatal exposure

## Abstract

Abdominal congenital malformations are responsible for early mortality, inadequate nutrient intake, and infant biological dysfunction. Exposure to metallic elements in utero is reported to be toxic and negatively impacts ontogeny. However, no prior study has sufficiently evaluated the effects of exposure to metallic elements in utero on abdominal congenital malformations. The aim of the present study was to evaluate associations between metallic elements detected in maternal blood during pregnancy and congenital abdominal malformations. Data from participants in the Japan Environment and Children’s Study was used in the present study, and contained information on singleton and live birth infants without congenital abnormalities (control: n = 89,134) and abdominal malformations (case: n = 139). Heavy metals such as mercury (Hg), lead (Pb), cadmium (Cd), and trace elements of manganese (Mn) and selenium (Se) were detected in maternal serum samples during mid- and late-gestation. Infant congenital abnormalities were identified from delivery records at birth or one month after birth by medical doctors. In a multivariate analysis adjusted to account for potential confounders, quartiles of heavy metals and trace elements present in maternal blood were not statistically correlated to the prevalence of abdominal congenital malformations at birth. This study is the first to reveal the absence of significant associations between exposure levels to maternal heavy metals and trace elements in utero and the prevalence of abdominal congenital malformations in a large cohort of the Japanese population. Further studies are necessary to investigate the impact of exposure to heavy metals and trace elements via maternal blood in offspring after birth.

## 1. Introduction

Congenital anomalies are responsible for 11% of neonatal mortality worldwide [1]. In Japan, congenital anomalies, including congenital malformations, morphological anomalies, and chromosome aberrations, have been the main causes of neonatal mortality since 1990 and are responsible for 40.3% of domestic neonatal mortality [2]. Some of the causes of congenital anomalies are chromosomal abnormalities, single gene disorders, maternal infection, and medications during pregnancy; however, the majority of the causes remain unclear.

Heavy metals such as lead (Pb), cadmium (Cd), mercury (Hg), and trace elements of selenium (Se) and manganese (Mn) are ingested by the mother through daily food intake, and inhalation by air pollutants that cross the placenta and reach their fetus. Maternal heavy metals and trace elements are associated with fetal cell differentiation and development, oxidative stress, and metabolic function. Prenatal exposure to heavy metals and trace elements may play a role as genetic toxicants in fetal immature organs and developing tissues that could relate to the development of congenital anomalies [3,4,5,6]. In the past, genetic toxicants in methylmercury poisoning accidents have been reported to have severe harmful effects on the fetus, including stillbirth, fetal growth retardation, and congenital malformations [7].

Several human studies have evaluated the risk of congenital heart defects, cleft lip and palate, and neural tube defects due to maternal exposure to heavy metals and trace elements [8]. However, only a limited number of human studies have targeted abdominal malformation. Abdominal congenital malformations are responsible for early mortality, inadequate nutrient intake, and biological dysfunction in infants. Previous studies on the topic have either not adequately assessed the concentrations of metals and trace elements in the maternal body or had small sample sizes, hindering effective comparisons between cases and controls. Therefore, at present, there is insufficient evidence linking the concentrations of metals and trace elements in maternal blood to abdominal congenital malformations.

The present study used data and biological samples from participants in the Japan Environment and Children’ s Study (JECS). The difference in metal and trace element concentrations between the JECS study participants, North Americans and Europeans can be attributed to maternal smoking and alcohol drinking behavior, body size, socio-economic status, and a food preference for rice and fish/seafood in East Asians. Blood Pb and Cd levels decrease with time; however, recent studies specifically investigating their effects on the health of human offspring are limited. In fact, the JECS has reported adverse effects of maternal Cd and Mn on early preterm birth and birth size [9,10]. The aim of the present study was to evaluate any associations between maternal heavy metals and congenital abdominal malformations.

## 2. Materials and Methods

### 2.1. Participants and Data Collection in JECS

The JECS is a nationwide and government-funded birth cohort study that aims to investigate the effect of the environment on child health and development. The study design and baseline profile of JECS participants have been previously described [11,12]. Briefly, in the recruitment period from January 2011 to March 2014, a total of 103,060 pregnancies were recorded, of which 100,304 were live births. Maternal self-administered questionnaires were completed at mothers’ first gestations, and once again during their mid- to late-gestations. The maternal questionnaires included maternal and paternal data such as smoking and drinking habits, paternal smoking habits, maternal and paternal educational duration, and household income. A total of 51,402 male partners provided paternal consent forms during their female partner’s pregnancy, and provided data on paternal age at registration, of which 46,929 had female partners who were delivering live births.

### 2.2. Ethical Statement

The JECS protocol was approved by the Ministry of the Environment’s Institutional Review Board on Epidemiological Studies and the ethics committees of each participating institution (Appendix A) (ethical project identification code: Kanken19–117). All participants gave written informed consent for inclusion before they participated in the study. The study was conducted in accordance with the Declaration of Helsinki (sixth version).

### 2.3. Identified Congenital Abnormalities

Details of the collection of delivery records, categorization, and prevalence of congenital malformations have been published elsewhere [13]. Briefly, delivery records at birth or one month after birth were collected by transcription of medical records, which included data regarding maternal pre-pregnancy weight, height, pregnancy complications, infant birth weight, height, sex, and congenital abnormalities. Infants’ congenital abnormalities were identified from the delivery records at birth or one month after birth by medical doctors at each co-operating health care provider. Abdominal congenital malformations included seven types of anomalies: omphalocele, gastroschisis, esophageal atresia with or without fistula, duodenal atresia, intestinal atresia, anorectal atresia, and diaphragmic hernia [13].

### 2.4. Measurements of the Blood Content of Five Metallic Elements

Maternal blood was collected from mid-late trimesters and stored at −80 °C until the time of analysis. Detailed information on the analytic process and quality control has been previously described elsewhere [14]. The blood content level of five metallic elements (ME), mercury (Hg), lead (Pb), cadmium (Cd), manganese (Mn) and selenium (Se), were measured using inductively coupled plasma mass spectrometry (ICP-MS) analyses of maternal serum samples from 96,696 participants.

The JECS protocol was approved by the Ministry of the Environment’s Institutional Review Board on Epidemiological Studies, and the ethics committees of all participating institutions, listed as follows: the National Institute for Environmental Studies, the National Center for Child Health and Development, Hokkaido University, Sapporo Medical University, Asahikawa Medical College, Japanese Red Cross Hokkaido College of Nursing, Tohoku University, Fukushima Medical University, Chiba University, Yokohama City University, University of Yamanashi, Shinshu University, University of Toyama, Nagoya City University, Kyoto University, Doshisha University, Osaka University, Osaka Medical Center and Research Institute for Maternal and Child Health, Hyogo College of Medicine, Tottori University, Kochi University, University of Occupational and Environmental Health, Kyushu University, Kumamoto University, University of Miyazaki, and University of the Ryukyu. The JECS was conducted in accordance with the Declaration of Helsinki and other nationally valid regulations. Written informed consent was obtained from all participants. The present study used the JECS dataset “jecs-ta-20190930,” which was released in October 2019. The present study also used the jecs-ag-20160424 dataset, which was released in June 2016 and revised in October 2016, along with the supplementary dataset, jecs-ag-20160424-sp1.

### 2.5. Selected Participants in Present Study

The flowchart in Figure 1 illustrates when the various types of data were collected from participants. Concentrations of ME in maternal blood from 96,696 participants in their mid- and late-trimesters were analyzed. Of these, 94,661 sets of participant data contained the following: self-administered questionnaires, birth records without missing data on maternal age, maternal smoking and drinking habits, paternal smoking habits, birth year of infant, and sex of infant. We excluded infants with chromosomal abnormalities, congenital malformations without abdominal anomalies, miscarriages, and those that were stillbirths or from multiple births, but included singleton and live birth infants without congenital abnormalities (control: n = 89,134) and abdominal malformations (n = 139).

### 2.6. Statistical Analysis

The differences in the concentrations of ME detected in maternal blood, parental characteristics, and infant characteristics between abdominal malformed infants and non-abdominal malformed infants were analyzed using Pearson’s chi-square tests, Student’s *t*-tests, Spearman’s rank correlation test, Mann–Whitney U-test, and Kruskal–Wallis test (Table 1, Table 2, Table 3 and Table 4).

The participants were divided into quartiles according to distributed concentrations of each of the five ME. In a multivariable logistic regression analysis, with abdominal malformation as the dependent variable, the odds ratio (OR) and 95% confidence interval (CI) of abdominal malformation for the second, third, and fourth quartiles of the concentrations of ME present in maternal blood, compared with the lowest quartile as reference, were calculated. To evaluate the dose–response relationship, a *p*-trend was calculated for the quartile groups of categorized concentrations of ME in maternal blood, as an ordinal variable (Figure 2, Table 5, Table 6 and Table 7). Outcomes of all abdominal malformations and each of the seven types of anomalies, including omphalocele, gastroschisis, esophageal atresia with or without fistula, duodenal atresia, intestinal atresia, anorectal atresia, and diaphragmic hernia were used in a multivariable logistic regression analysis, adjusted for potential confounders. These confounders comprised factors from the present study or those previously reported to be significantly associated with maternal ME and abdominal congenital malformations, such as maternal age at delivery (categorical) and maternal smoking habit (never, quit, smoking during pregnancy), drinking habit (never, quit, drinking during pregnancy), paternal smoking habit (never, quit, smoking during pregnancy), birth year of infants, and sex of infants.

Statistical significance was set at *p* < 0.05. Statistical analyses were performed using SPSS software for Windows (version 21.0J; IBM, Armonk, NY, USA).

## 3. Results

### Parental Characteristics

For 89,273 participants in the present study, the median concentrations of maternal ME, including Pb, Cd, Hg, Se, and Mn, were 5.84 ng/g, 0.661 ng/g, 3.64 ng/g, 168.0 ng/g, and 15.3 ng/g, respectively. There were no significant differences in the concentrations of maternal ME between infants with and without malformation (Table 1). Abdominal congenital malformations were observed in 139 infants, included in the current study by transcription of medical records at birth or one month after birth (Table 2). Parental and infant characteristics are shown in Table 3 and Table 4, respectively. The prevalence of abdominal congenital malformations significantly differed according to maternal smoking habit, type of delivery, sex of infants, gestational age, and birth weight, but not maternal age, drinking habit, educational duration, and household income (Table 3 and Table 4). Concentrations of maternal ME were significantly different between maternal, paternal, and infant characteristics.

Logistic regression analysis revealed no statistically significant (*p* < 0.05) associations between maternal ME and any abdominal congenital malformations (Figure 2 and Table 5). In relation to specific abdominal congenital malformations, compared with the lowest quartile (*p*-trend = 0.089), the third quartile of Hg, adjusted for ORs of diaphragmatic hernia, was significantly increased (OR (95% CI): 3.22 (1.05, 9.90)) (Table 6 and Table 7). A significant inverse trend was observed for diaphragmatic hernia throughout the quartiles of Mn (*p*-trend = 0.010). The highest (OR (95% CI): 0.15 (0.03, 0.68)) of Mn, adjusted for ORs of diaphragmatic hernia, were significantly decreased, compared with the lowest quartiles (Table 6 and Table 7). A significant inverse trend was observed for omphalocele throughout the quartiles of Pb (*p*-trend = 0.033). The second (OR (95% CI): 9.96 (1.27, 77.88)) and third quartiles (OR (95% CI): 10.06 (1.28, 78.77)) of Se, adjusted for ORs of omphalocele, were significantly increased, compared with the lowest quartiles (Table 6 and Table 7). We found no significant associations between maternal ME and specific abdominal congenital malformations, including gastroschisis, esophageal atresia with or without fistula, duodenal atresia/stenosis, intestinal atresia/stenosis, and anorectal atresia/stenosis (Table 6 and Table 7).

**Table 5 ijerph-18-10103-t005:** Odds ratios for all abdominal congenital malformations according to the concentrations of metallic elements detected in maternal blood at mid-late trimester.

Cd	Case /Control (n)	Adjusted OR (95% CI)	Pb	Case /Control (n)	Adjusted OR (95% CI)	Hg	Case /Control (n)	Adjusted OR (95% CI)	Se	Case /Control (n)	Adjusted OR (95% CI)	Mn	Case /Control (n)	Adjusted OR (95% CI)
Q1	34/23,117	1	Q1	37/22,571	1	Q1	37/22,268	1	Q1	36/22,348	1	Q1	40/22,507	1
Q2	44/21,323	1.46 (0.93, 2.29)	Q2	44/22,174	1.19 (0.76, 1.84)	Q2	29/22,218	0.80 (0.49, 1.31)	Q2	34/22,717	0.93 (0.58, 1.49)	Q2	35/22,858	0.86 (0.55, 1.36)
Q3	26/22,951	0.80 (0.48, 1.35)	Q3	28/22,382	0.77 (0.47, 1.26)	Q3	41/22,335	1.13 (0.72, 1.77)	Q3	41/22,889	1.11 (0.71, 1.74)	Q3	33/22,080	0.83 (0.52, 1.32)
Q4	35/21,743	1.22 (0.75, 2.00)	Q4	30/22,007	0.85 (0.52, 1.38)	Q4	32/22,313	0.88 (0.55, 1.42)	Q4	28/21,180	0.80 (0.48, 1.32)	Q4	31/21,689	0.78 (0.48, 1.25)
*p* for trend		0.997			0.233			0.963			0.585			0.296

OR: odds ratio was adjusted for maternal age, smoking habit, drinking habit, paternal smoking habit, birth year of child, and sex of the child. Q1: first quartile, Q2: second quartile, Q3: third quartile, Q4: fourth quartile, CI: confidence interval.

**Table 6 ijerph-18-10103-t006:** Odds ratios for each abdominal congenital malformation (Diaphragmatic hernia, Omphalocele, and Gastroschisis) according to the concentrations of metallic elements detected in maternal blood at mid-late trimester.

Cd	Case/ Control (n)	Adjusted OR (95% CI)	Pb	Case/ Control (n)	Adjusted OR (95% CI)	Hg	Case/ Control (n)	Adjusted OR (95% CI)	Se	Case/ Control (n)	Adjusted OR (95% CI)	Mn	Case/ Control (n)	Adjusted OR (95% CI)
Diaphragmatic hernia														
Q1	10/23,117	1	Q1	9/22,571	1	Q1	4/22,268	1	Q1	7/22,348	1	Q1	13/22,507	1
Q2	11/21,323	1.24 (0.52, 2.93)	Q2	11/22,174	1.24 (0.51, 2.99)	Q2	8/222,18	2.00 (0.60, 6.64)	Q2	7/22,717	0.95 (0.33, 2.71)	Q2	11/22,858	0.80 (0.36, 1.79)
Q3	10/22,951	1.08 (0.44, 2.62)	Q3	8/22,382	0.89 (0.34, 2.31)	Q3	13/22,335	3.22 (1.05, 9.90) *	Q3	9/22,889	1.18 (0.44, 3.18)	Q3	9/22,080	0.68 (0.29, 1.60)
Q4	4/21,743	0.48 (0.15, 1.58)	Q4	7/22,007	0.81 (0.30, 2.20)	Q4	10/22,313	2.50 (0.78, 8.00)	Q4	12/21,180	1.68 (0.66, 4.31)	Q4	2/21,689	0.15 (0.03, 0.68) *
*p* for trend		0.297			0.543			0.089			0.221			0.010
Omphalocele														
Q1	7/23,117	1	Q1	11/22,571	1	Q1	7/22,268	1	Q1	1/22,348	1	Q1	7/22,507	1
Q2	8/21,323	1.26 (0.45, 3.51)	Q2	8/22,174	0.72 (0.29, 1.81)	Q2	3/22,218	0.44 (0.11, 1.71)	Q2	10/22,717	9.96 (1.27, 77.88) *	Q2	8/22,858	1.14 (0.41, 3.16)
Q3	3/22,951	0.44 (0.11, 1.74)	Q3	4/22,382	0.35 (0.11, 1.12) †	Q3	10/22,335	1.50 (0.57, 3.96)	Q3	10/22,889	10.06 (1.28, 78.77) *	Q3	4/22,080	0.59 (0.17, 2.03)
Q4	9/21,743	1.42 (0.50, 4.00)	Q4	4/22,007	0.35 (0.11, 1.13) †	Q4	7/22,313	1.10 (0.38, 3.15)	Q4	6/21,180	6.56 (0.78, 54.75) †	Q4	8/21,689	1.22 (0.44, 3.37)
*p* for trend		0.846			0.033			0.439			0.152			0.962
Gastroschisis														
Q1	4/23,117	1	Q1	0/22,571	-	Q1	2/22,268	1	Q1	4/22,348	1	Q1	2/22,507	1
Q2	2/21,323	0.55 (0.10, 3.03)	Q2	2/22,174	1	Q2	4/22,218	2.06 (0.38, 11.26)	Q2	1/22,717	0.24 (0.03, 2.13)	Q2	4/22,858	1.92 (0.35, 10.49)
Q3	0/22,951	-	Q3	2/22,382	1.00 (0.14, 7.09)	Q3	1/22,335	0.53 (0.05, 5.89)	Q3	3/22,889	0.72 (0.16, 3.25)	Q3	2/22,080	0.97 (0.14, 6.91)
Q4	3/21,743	0.88 (0.18, 4.28)	Q4	5/22,007	2.63 (0.50, 13.70)	Q4	2/22,313	1.07 (0.15, 7.71)	Q4	1/21,180	0.26 (0.03, 2.34)	Q4	1/21,689	0.47 (0.04, 5.26)
*p* for trend		0.935			0.212			0.717			0.317			0.431

OR: odds ratio was adjusted for maternal age, smoking habit, drinking habit, paternal smoking habit, birth year of child, and sex of the child. Q1: first quartile, Q2: second quartile, Q3: third quartile, Q4: fourth quartile, CI: confidence interval. † *p* < 0.1, * *p* < 0.05. - no cases of abdominal malformation (number of case = 0), rather than OR value.

**Table 7 ijerph-18-10103-t007:** Odds ratios for each abdominal congenital malformation (Esophageal atresia with or without fistula, Duodenal atresia/stenosis, Intestinal atresia/stenosis, and Anorectal atresia/stenosis) according to the concentrations of metallic elements detected in maternal blood at mid-late trimester.

Cd	Case/ Control (n)	Adjusted OR (95% CI)	Pb	Case/ Control (n)	Adjusted OR (95% CI)	Hg	Case/ Control (n)	Adjusted OR (95% CI)	Se	Case/ Control (n)	Adjusted OR (95% CI)	Mn	Case/ Control (n)	Adjusted OR (95% CI)
Esophageal atresia with or without fistula														
Q1	3/23,117	1	Q1	2/22,571	1	Q1	3/22,268	1	Q1	2/22,348	1	Q1	3/22,507	1
Q2	6/21,323	2.94 (0.59, 14.72)	Q2	2/22,174	0.49 (0.04, 5.43)	Q2	1/22,218	0.48 (0.04, 5.31)	Q2	3/22,717	1.67 (0.26, 10.97)	Q2	3/22,858	1.45 (0.24, 8.69)
Q3	0/22,951	-	Q3	2/22,382	0.95 (0.13, 6.80)	Q3	3/22,335	1.39 (0.23, 8.36)	Q3	2/22,889	1.06 (0.14, 8.21)	Q3	4/22,080	1.98 (0.36, 10.84)
Q4	1/21,743	0.44 (0.04, 5.10)	Q4	4/22,007	1.88 (0.33, 10.50)	Q4	3/22,313	1.36 (0.23, 8.27)	Q4	3/21,180	1.29 (0.19, 8.84)	Q4	0/21,689	-
*p* for trend		0.362			0.346			0.527			0.946			0.424
Duodenal atresia/stenosis														
Q1	3/23,117	1	Q1	4/22,571	1	Q1	3/22,268	1	Q1	5/22,348	1	Q1	0/22,507	-
Q2	2/21,323	0.72 (0.12, 4.39)	Q2	1/22,174	0.25 (0.03, 2.27)	Q2	3/22,218	0.99 (0.20, 4.90)	Q2	2/22,717	0.39 (0.08, 2.02)	Q2	3/22,858	1
Q3	3/22,951	0.98 (0.19, 5.05)	Q3	2/22,382	0.50 (0.09, 2.75)	Q3	2/22,335	0.64 (0.11, 3.82)	Q3	4/22,889	0.76 (0.20, 2.86)	Q3	4/22,080	1.37 (0.31, 6.14)
Q4	3/21,743	1.01 (0.19, 5.44)	Q4	4/22,007	0.99 (0.24, 4.06)	Q4	3/22,313	0.92 (0.18, 4.62)	Q4	0/21,180	-	Q4	4/21,689	1.35 (0.30, 6.04)
*p* for trend		0.915			0.910			0.812			0.666			0.702
Intestinal atresia/stenosis														
Q1	3/23,117	1	Q1	3/22,571	1	Q1	3/22,268	1	Q1	2/22,348	1	Q1	4/22,507	1
Q2	6/21,323	2.28 (0.56, 9.20)	Q2	4/22,174	1.40 (0.31, 6.29)	Q2	4/22,218	1.30 (0.29, 5.81)	Q2	5/22,717	2.41 (0.47, 12.47)	Q2	2/22,858	0.47 (0.09, 2.58)
Q3	2/22,951	0.73 (0.12, 4.46)	Q3	3/22,382	1.06 (0.21, 5.27)	Q3	4/22,335	1.26 (0.28, 5.64)	Q3	6/22,889	2.82 (0.56, 14.06)	Q3	2/22,080	0.49 (0.09, 2.68)
Q4	2/21,743	0.87 (0.14, 5.46)	Q4	3/22,007	1.12 (0.22, 5.64)	Q4	2/22,313	0.62 (0.10, 3.76)	Q4	0/21,180	-	Q4	5/21,689	1.22 (0.33, 4.57)
*p* for trend		0.575			0.989			0.643			0.215			0.717
Anorectal atresia/stenosis														
Q1	5/23,117	1	Q1	10/22,571	1	Q1	11/22,268	1	Q1	15/22,348	1	Q1	12/22,507	1
Q2	11/21,323	2.57 (0.89, 7.43) †	Q2	15/22,174	1.65 (0.74, 3.67)	Q2	6/22,218	0.54 (0.20, 1.47)	Q2	7/22,717	0.49 (0.20, 1.19)	Q2	5/22,858	0.41 (0.14, 1.15) †
Q3	9/22,951	2.02 (0.67, 6.11)	Q3	5/22,382	0.57 (0.19, 1.68)	Q3	10/22,335	0.90 (0.38, 2.12)	Q3	7/22,889	0.50 (0.20, 1.24)	Q3	7/22,080	0.58 (0.23, 1.47)
Q4	10/21,743	2.65 (0.88, 8.00) †	Q4	5/22,007	0.62 (0.21, 1.83)	Q4	8/22,313	0.69 (0.28, 1.73)	Q4	6/21,180	0.48 (0.18, 1.24)	Q4	11/21,689	0.93 (0.41, 2.11)
*p* for trend		0.149			0.158			0.633			0.099			0.969

OR: odds ratio was adjusted for maternal age, smoking habit, drinking habit, paternal smoking habit, birth year of child, and sex of the child. Q1: first quartile, Q2: second quartile, Q3: third quartile, Q4: fourth quartile, CI: confidence interval. † *p* < 0.1; no cases of abdominal malformation (number of case = 0), rather than OR value.

## 4. Discussion

We found no statistically significant associations between maternal ME and any abdominal congenital malformations. However, significant positive or inverse associations were observed among maternal Pb, Mn, Hg, and Se and specific abdominal congenital malformations, including diaphragmatic hernia and omphalocele, with a wide range in 95% CI, due to the small sample size. The results indicated that presence of maternal ME was not a definite risk factor for congenital malformation in the offspring. This is the first study to show that exposure to ME via maternal blood is unlikely to cause fetal abdominal congenital malformations.

Direct comparison of results between previous studies and the present study is difficult, because there are no previous studies assessing the effects of ME in maternal blood. In North Carolina, a human study was conducted to evaluate the association between metal concentrations in private well water and birth defect prevalence, including heat defects, cleft palate/lip, hypospadias, pyloric stenosis, and gastroschisis. Individual exposure was designated as the average metal concentrations in the census tract encompassing the geocoded maternal residence. No association was observed between abdominal congenital malformation and Mn and Cd, excluding the inverse association between Cd and prevalence of pyloric stenosis (RR: 0.4 95% CI: 0.3–0.7) [15]. In another study, based on the classification of exposure to emissions from municipal solid waste incinerators, data from exposed (n = 194) and unexposed (n = 2678) settlement populations was used to evaluate the relative risks of congenital malformations. The rate of congenital anomalies, including diaphragmatic hernia and gastroschisis, was not significantly higher in exposed than unexposed communities [16]. In a recent study, metal concentrations in maternal blood from 140 pregnant women living in industrial areas were measured; however, Cd was detected in urine samples from only two pregnant women who birthed infants without congenital anomalies (detection rate: 1.4%, median: 5.3 µg/L), which means that the effect on congenital anomalies could not be effectively statistically evaluated [17]. While the three afore-mentioned studies described inadequate and weak effects of ME, our present study provides more tangible results contending that maternal exposure to ME via maternal blood (at levels detected in this study) are unlikely to cause fetal abdominal congenital malformation.

Adjusted ORs for infant diaphragmatic hernia and omphalocele increased the upper Hg and Se. The potential biological mechanisms are that Hg can disturb cell differentiation and growth, and Se can affect fetal growth, because the normal range of Se is very narrow. Moreover, previously reported exposure concentrations of Hg among JECS participants estimated that almost 30% of the participants had blood Hg levels within the range where detrimental health effects cannot be excluded with adequate certainty [14]. On the other hand, we cannot exclude the possibility that this result was due to chance, because there is no evidence of the toxicity of Hg and Se on the specific biological mechanisms of diaphragmatic hernia and omphalocele, gastroschisis, esophageal atresia with or without fistula, duodenal atresia/stenosis, intestinal atresia/stenosis, or anorectal atresia/stenosis. Conversely, the ORs of infantile omphalocele and diaphragmatic hernia decreased in the upper Pb and Mn. A previous JECS study reported a positive correlation between maternal Pb and abstinence from alcohol during pregnancy. Another study among Japanese women reported that the major contribution rates to the total estimated uptake of Mn were Japanese tea (33.9%), rice (23.7%), and vegetables (13.6%) [18]. Therefore, our study suggests that Pb and Mn concentrations in maternal blood were at levels that did not negatively impact fetal development, and that high-quality nutritional intake may safeguard against anomalous fetal development.

In a previous JECS study on congenital anomalies, which defined an index scaling of occurrence per 10,000 pregnancies, prevalence for diaphragmatic hernia was 1.9, omphalocele was 2.6, gastroschisis was 1.5, esophageal atresia with or without fistula was 1.5, and duodenal atresia/stenosis was 1.1, intestinal atresia/stenosis was 0.8, and anorectal atresia/stenosis was 2.9 [13]. Applying the same index scaling to the present study targeting infants who were singleton and live births, the prevalence of abdominal congenital malformations was lower in all participants [13]. In the present study, malformed abdominal infants had lower gestational ages and birth weights than non-malformed infants (Table 4), which is consistent with reports from previous human studies [19]. Congenital anomalies begin to develop in the early onset of fetal development and involve risk factors common to structural abnormalities and fetal growth retardation, such as prenatal complications, chromosomal anomalies, placental insufficiency, and polyhydramnios [19]. Maternal smoking is a known risk factor for birth defects [20]. However, in the present study, mothers with malformed abdominal infants had a lower frequency of smoking during pregnancy and a higher frequency of quitting smoking before pregnancy than those without abdominal malformed infants. The results suggest that maternal smoking during pregnancy cannot be responsible for abdominal malformed infants, and mothers may behave in a protective manner according to maternal awareness of self-risk of birth defects such as older childbearing age. Previously, associations between maternal alcohol consumption and malformed abdominal infants were weak [21]. In the present study, maternal alcohol consumption during pregnancy was not associated with malformed abdominal infants, a finding consistent with previous studies that reported that maternal alcohol consumption during pregnancy has a weak effect on the risk of cryptorchidism [22] and congenital heart defects [23].

## 5. Strengths and Limitations

The strengths of this study are that basic data on early gestations in participants have been prospectively collected using a national cohort study, which had a large sample size and included a sizable geographic proportion of Japan. ME in maternal blood were detected at a comprehensive detection rate. Cases of abdominal malformation were identified from hospital records by the obstetrician, which minimalized miscalculation bias. Our study did have some limitations. Firstly, we have no data on the history of abdominal congenital malformation in participating parents. Unobserved potential confounders may also be associated with maternal ME and abdominal malformations. Further studies are needed to evaluate the effect of exposure to maternal ME in utero on offspring health outcomes, including growth and development.

## 6. Conclusions

To our knowledge, this study is the first to reveal no significant associations between exposure levels in utero of maternal ME such as Hg, Pb, Cd, Mn, and Se, and the prevalence of abdominal congenital malformations in a large cohort of the Japanese population. However, there is a possibility that maternal ME may indeed have adverse effects on fetal development and subsequent health in children. Further studies with follow-up of children are necessary to investigate the effects of maternally present heavy metals on offspring after birth.

## Figures and Tables

**Figure 1 ijerph-18-10103-f001:**
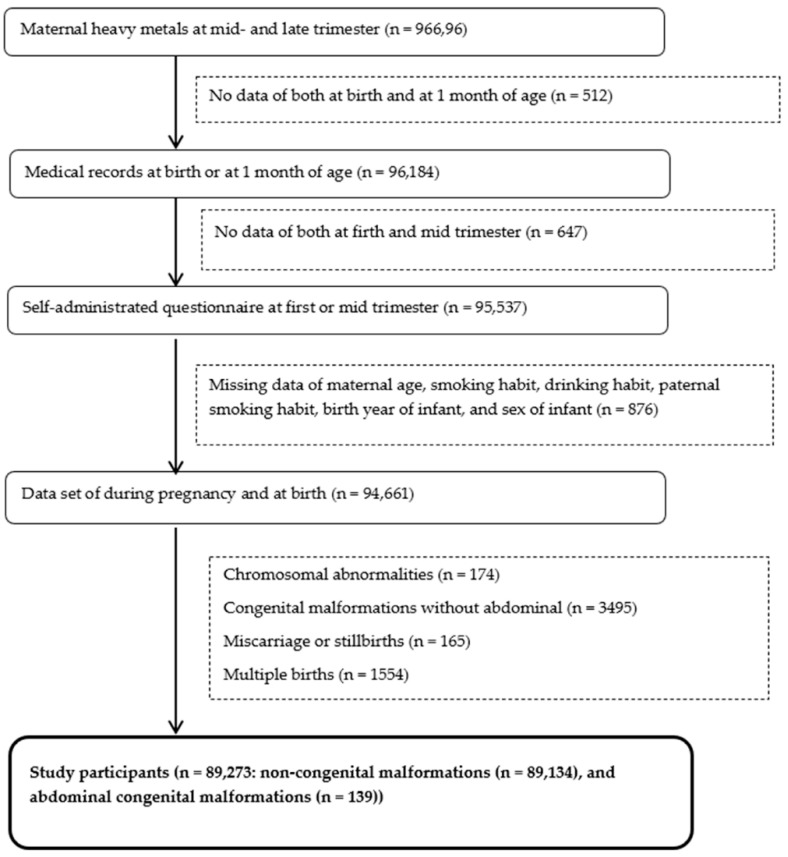
Flowchart of participant data-collection procedures.

**Figure 2 ijerph-18-10103-f002:**
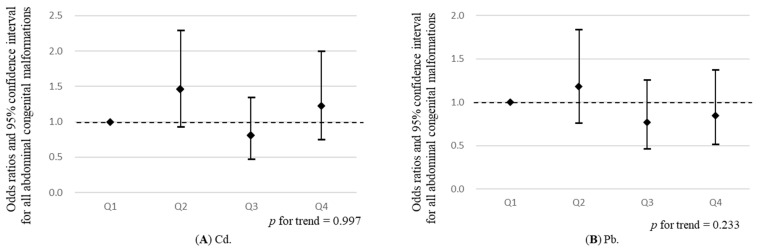
Associations between the quartiles of maternal metallic elements (mercury (Hg), lead (Pb), cadmium (Cd), manganese (Mn), and selenium (Se)) (*x*-axes), and odds ratios and 95% confidence interval for all abdominal congenital malformations (*y*-axes), compared with the Q1, in a multivariable logistic regression analysis, adjusted for maternal age at delivery and maternal smoking habit, drinking habit, paternal smoking habit, birth year of infants, and sex of infants. Trend correlation was calculated using quartile group of maternal metallic elements, which were analyzed as ordinal variables. Q1: first quartile, Q2: second quartile, Q3: third quartile, Q4: fourth quartile. (**A**), (**B**), (**C**), (**D**) and (**E**) show associations between all abdominal congenital malformations and metallic elements; Cd, Pb, Hg, Se, and Mn, respectively.

**Table 1 ijerph-18-10103-t001:** Concentrations of metallic elements in maternal blood at mid-late trimester.

Metallic Elements	All (n = 89,273)					Normal (n = 89,134)	Abdominal Congenital Malformations (n = 139)	*p*
	Minimum	25th Percentile	Median	75th Percentile	Maximum	Median (IQR)	Median (IQR)	
Pb (ng/g)	1.2	4.7	5.84	7.32	110	5.85 (4.70, 7.32)	5.53 (4.59, 7.00)	0.094
Cd (ng/g)	0.0951	0.494	0.661	0.902	5.33	0.66 (0.49, 0.90)	0.62 (0.50, 0.92)	0.546
Hg (n/g)	0.182	2.55	3.64	5.2	58.8	3.64 (2.55, 5.20)	3.69 (2.45, 5.01)	0.827
Se (ng/g)	82.8	156	168	182	976	168.00 (156.00, 182.00)	168.00 (155.00, 180.00)	0.773
Mn (ng/g)	3.06	12.6	15.3	18.6	60.8	15.30 (12.60, 18.60)	14.90 (12.20, 18.20)	0.458

*p* by Mann–Whitney U test, with significance at *p* < 0.05. IQR: interquartile range.

**Table 2 ijerph-18-10103-t002:** Cases of congenital abdominal malformations.

Abdominal Congenital Malformations	ICD-10 Code	All	Male	Female	Unknown
		N	N (%)	N (%)	N (%)
Diaphragmatic hernia	Q79.0	35	16 (45.7)	19 (54.3)	0 (0.0)
Omphalocele	Q79.2	27	11 (40.7)	16 (59.3)	0 (0.0)
Gastroschisis	Q79.3	9	6 (66.7)	3 (33.3)	0 (0.0)
Esophageal atresia with or without fistula	Q39.0 Q39.1	10	5 (50.0)	4 (40.0)	1 (10.0)
Duodenal atresia/stenosis	Q41.0	11	7 (63.6)	4 (36.4)	0 (0.0)
Intestinal atresia/stenosis	Q41.1–Q41.9	13	6 (46.2)	7 (53.8)	0 (0.0)
Anorectal atresia/stenosis	Q42.0–Q42.9	35	21 (60.0)	14 (40.0)	0 (0.0)

ICD-10 code: Transcriptions of both the ICD-10 code and congenital anomaly names in medical records were collected during the data collection phase.

**Table 3 ijerph-18-10103-t003:** Parental characteristics for births with/without abdominal congenital malformations.

Characteristics	All		Non-Malformed	Abdominal Congenital Malformations
Maternal Characteristics				
Age at delivery (years old)	N or N (%)	Mean ± SD	N (%)	N (%)
<25	10,265 (11.50)		10,244 (11.49)	21 (15.11)
25 to <30	26,545 (29.73)		26,507 (29.74)	38 (27.34)
30 to <35	31,260 (35.02)		31,216 (35.02)	44 (31.65)
≥35	21,203 (23.75)		21,167 (23.75)	36 (25.90)
			Mean ± SD	Mean ± SD
Pre-pregnancy weight	89,234	53.12 ± 8.85	53.11 ± 8.85	53.59 ± 9.01
Height	89,253	158.12 ± 5.35	158.12 ± 5.35	158.57 ± 4.95
Parity	N or N (%)		N (%)	N (%)
Nulliparous	43,097 (48.44)		43,022 (48.43)	75 (54.35)
Multiparous	45,870 (51.56)		45,807 (51.57)	63 (45.65)
Smoking habit				
Never	51,924 (58.16)		51,846 (58.17)	78 (56.12)
Quit before pregnancy	21,217 (23.77)		21,172 (23.75)	45 (32.37)
Smoking during pregnancy	16,132 (18.07)		16,116 (18.08)	16 (11.51)
Drinking habit				
Never	30,825 (34.53)		30,773 (34.52)	52 (37.41)
Quit before pregnancy	49,330 (55.26)		49,252 (55.26)	78 (56.12)
Drinking during pregnancy	9118 (10.21)		9109 (10.22)	9 (6.47)
Educational duration (year)				
<10	4137 (4.69)		4129 (4.68)	8 (5.84)
10–<13	27,892 (31.59)		27,855 (31.60)	37 (27.01)
13–<15	37,288 (42.23)		37,218 (42.22)	70 (51.09)
15–<17	18,977 (21.49)		18,955 (21.50)	22 (16.06)
Paternal characteristics				
Age at entry (years old)	N or N (%)		N (%)	N (%)
<25	3385 (7.22)		3379 (7.22)	6 (8.57)
25 to <30	11,210 (23.92)		11,194 (23.92)	16 (22.86)
30 to <35	15,516 (33.11)		15,485 (33.09)	31 (44.29)
≥35	16,753 (35.75)		16,736 (35.77)	17 (24.29)
Smoking habit				
Never	24,084 (26.98)		24,051 (26.98)	33 (23.74)
Quit before pregnancy	20,753 (23.25)		20,713 (23.24)	40 (28.78)
Smoking during pregnancy	44,436 (49.78)		44,370 (49.78)	66 (47.48)
Educational duration (year)				
<10	6430 (7.31)		6423 (7.31)	7 (5.11)
10–<13	32,538 (36.98)		32,489 (36.99)	49 (35.77)
13–<15	19,880 (22.60)		19,850 (22.60)	30 (21.90)
15–<17	29,132 (33.11)		29,081 (33.11)	51 (37.23)
Household income (million yen)				
<2	4622 (5.59)		4614 (5.59)	8 (6.30)
2–<4	28,705 (34.73)		28,663 (34.74)	42 (33.07)
4–<6	27,334 (33.08)		27,290 (33.07)	44 (34.65)
6–<8	13,115 (15.87)		13,092 (15.87)	23 (18.11)
8–<10	5359 (6.48)		5351 (6.48)	8 (6.30)
≥10	3507 (4.24)		3505 (4.25)	2 (1.57)

SD, standard deviation.

**Table 4 ijerph-18-10103-t004:** Infant characteristics for births with/without abdominal congenital malformations.

	All		Non-Malformed	Abdominal Congenital Malformations	*p*
Infant characteristics					
	N or N (%)	Mean ± SD	Mean ± SD	Mean ± SD	*p* ^a^
Gestational weeks	89235	38.87 ± 1.46	38.87 ± 1.46	37.76 ± 2.68	<0.001
Birth weight	89183	3033.77 ± 402.47	3034.21 ± 401.95	2743.65 ± 594.58	<0.001
			N (%)	N (%)	*p* ^b^
Vaginal delivery	72,701 (81.65)		72,612 (81.68)	89 (64.96)	<0.001
Cesarean section	16,339 (18.35)		16,291 (18.32)	48 (35.04)	
Sex					
Male	45,662 (51.15)		45,591 (51.15)	71 (51.08)	<0.001
Female	43,609 (48.85)		43,542 (48.85)	67 (48.20)	
Unknown	2 (0.00)		1 (0.00)	1 (0.72)	
Birth year					
2011	8713 (9.76)		8696 (9.76)	17 (12.23)	0.807
2012	25,335 (28.38)		25,297 (28.38)	38 (27.34)	
2013	31,815 (35.64)		31,767 (35.64)	48 (34.53)	
2014	23,410 (26.22)		23,374 (26.22)	36 (25.90)	

SD, standard deviation. *p* ^a^; T test, *p*
^b^; χ2 test.

## Data Availability

Data are available on reasonable request. Data are unsuitable for public deposition due to ethical restrictions and legal framework of Japan. It is prohibited by the Act on the Protection of Personal Information (Act No. 57 of 30 May 2003, amendment on 9 September 2015) to publicly deposit the data containing personal information. Ethical Guidelines for Medical and Health Research Involving Human Subjects enforced by the Japan Ministry of Education, Culture, Sports, Science and Technology and the Ministry of Health, Labour and Welfare also restricts the open sharing of the epidemiologic data. All inquiries about access to data should be sent to: jecs-en@nies.go.jp. The person responsible for handling enquiries sent to this email address is Dr Shoji F. Nakayama, JECS Programme Office, National Institute for Environmental Studies.

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
