# Peer review of "Association between the Concentrations of Metallic Elements in Maternal Blood during Pregnancy and Prevalence of Abdominal Congenital Malformations: The Japan Environment and Children’s Study"

_ijerph, 2021, doi:10.3390/ijerph181910103_

Round 1

Reviewer 1 Report

The manuscript presents an interesting topic. In my opinion, the design of the study and general description is appropriate.  I have some issues:

  1. In the abstract Authors wrote that „This is the first study to show that exposure to heavy metals and trace elements as detected in maternal blood, cannot cause in utero fetal abdominal congenital malformations”, I can not agree with this sentence. Based on your results you can conclude that there is no association between ME exposure and congenital malformations…and also in the conclusion Autors wrote, „However, there is a possibility that maternal ME may indeed have adverse effects on fetal development (…) .” In the abstract and in the text there is exclusive information. The conclusion should be changed.
  2. Table 5 is too long and not clear. In this table, „non-malformed’ group should be included.
  3. Results partly should be shown on Figures for more clear, especially table 5 is not clear enough.

Author Response

Revised 1: Response to Reviewer 1

Comment 1: In the abstract Authors wrote that „This is the first study to show that exposure to heavy metals and trace elements as detected in maternal blood, cannot cause in utero fetal abdominal congenital malformations”, I can not agree with this sentence. Based on your results you can conclude that there is no association between ME exposure and congenital malformations…and also in the conclusion Autors wrote, „However, there is a possibility that maternal ME may indeed have adverse effects on fetal development (…) .” In the abstract and in the text there is exclusive information. The conclusion should be changed.

Response 1: According to the reviewers' comments, we have changed the sentence from “This is the first study to show that exposure to heavy metals and trace elements as detected in maternal blood, cannot cause in utero fetal abdominal congenital malformations” to “This study is the first to reveal the absence of significant associations between exposure levels of maternal heavy metals and trace elements in utero and the prevalence of abdominal congenital malformations in a large cohort of the Japanese population.” in the abstract section. (p 1 line 37–40)

Comment 2: Table 5 is too long and not clear. In this table, „non-malformed’ group should be included.

Response 2: According to the reviewers' comments, we have changed Table 5 to new Tables 5 and 6. The layouts of the new tables (Tables 5 and 6) have been changed to the quartile groups of metallic elements within an outcome (abdominal malformation), which are in rows, and different metallic elements are in the columns. However, the resulting numbers remain unchanged.

Subsequently, we have revised sentences about quartile groups in the Materials and Methods section, to clearly explain the results targeting both abdominal malformed infants and non-abdominal malformed infants in Tables 5 and 6, as follows.

The differences in the concentrations of ME detected in maternal blood, parental characteristics, and infant characteristics between abdominal malformed infants and non-abdominal malformed infants were analyzed using Pearson's chi-square tests, Student’s t-tests, Spearman’s rank correlation test, Mann–Whitney U-test, and Kruskal–Wallis test (Tables 1, 2, 3, and 4). The participants were divided into quartiles according to distributed concentrations of each of the five ME. In a multivariable logistic regression analysis with abdominal malformation as the dependent variable, the odds ratio (OR) and 95% confidence interval (CI) of abdominal malformation for the second, third, and fourth quartiles of the concentrations of ME present in maternal blood, compared with the lowest quartile as reference, were calculated. To evaluate the dose-response relationship, a p-trend was calculated for the quartile groups of ME in maternal blood, as an ordinal variable (Figure 2, Tables 5 and 6).(p 4–5, line 152–163)

Response: However, when no cases of abdominal malformation were observed in one among the quartile groups, OR cannot be calculated in that group. Tables 5 and 6 indicate each number of participates with case / control, when the case = 0, indicated “ - “ rather than OR value.

We have changed “case/control” to “case/control (n)” in Tables 5 and 6.

We have added sentences in the footnote of Table 6, as follows:

-; no cases of abdominal malformation (number of case = 0), rather than OR value.

Comment 3: Results partly should be shown on Figures for more clear, especially table 5 is not clear enough.

Response 3: According to the reviewers' comments, we have added a new Figure 2 (p 9), illustrating the results of a multivariable logistic regression analysis with the quartiles of ME as the independent variable and the existence of all abdominal malformation as the dependent variable.

We have added explanation in the figure legend in Figure 2, as follows:

“Figure. 2 Associations between the quartiles of maternal metallic elements (mercury [Hg], lead [Pb], cadmium [Cd], manganese [Mn], and selenium [Se]) (X-axes), and odds ratios and 95% confidence interval for all abdominal congenital malformations (Y-axes), compared with the Q1, in a multivariable logistic regression analysis, adjusted for maternal age at delivery and maternal smoking habit, drinking habit, paternal smoking habit, birth year of infants, and sex of infants. Trend correlation was calculated using quartile group of maternal metallic elements, which were analyzed as ordinal variables. Q1: first quartile, Q2: second quartile, Q3: third quartile, Q4: fourth quartile.” (p 9, line 267–272)

Reviewer 2 Report

The paper entitled “Association between the concentrations of metallic elements in 2 maternal blood during pregnancy and prevalence of abdominal 3 congenital malformations: The Japan Environment and Chil-4 dren’s Study” is very well structured and presented. The main topics of the paper are well introduced and the methodology and results are well organized. This work is an important contribute to clarify the impact of heavy metals in congenital malformations.

Author Response

Response to Reviewer 2

Comment: The paper entitled “Association between the concentrations of metallic elements in 2 maternal blood during pregnancy and prevalence of abdominal 3 congenital malformations: The Japan Environment and Chil-4 dren’s Study” is very well structured and presented. The main topics of the paper are well introduced and the methodology and results are well organized. This work is an important contribute to clarify the impact of heavy metals in congenital malformations.

Response: Thank you for your comments. We are thankful for the time and effort you expended in the review.